# System for Monitoring the Safety and Movement Mechanics of Users of Bicycles and Electric Scooters in Real Conditions in the Context of Social Sustainability

Jakub Majer [1,2,†] , Jarosław Adamiec [2,*,†] , Maciej Obst [3,†] and Dariusz Kurpisz [3,†]

1   Łukasiewicz Research Network—Poznan Institute of Technology, Ewarysta Estkowskiego 6, 61-755 Poznan, Poland; jakub.majer@pit.lukasiewicz.gov.pl
2   Institute of Machine Design, Poznan University of Technology, Piotrowo 3, 60-965 Poznan, Poland
3   Institute of Applied Mechanics, Poznan University of Technology, Jana Pawla II 24, 60-965 Poznan, Poland; maciej.obst@put.poznan.pl (M.O.); dariusz.kurpisz@put.poznan.pl (D.K.)
*   Correspondence: jaroslaw.adamiec@put.poznan.pl; Tel.: +48-61-665-2054
†   These authors contributed equally to this work.

**Abstract:** Sustainable development means taking care of the environment, which also means promoting green transport, which involves the systematic development of personal transport in its broadest sense. The positive aspects associated with cheap and convenient electric transport are intertwined with the problem of collisions and accidents. While developing road infrastructure for electric vehicles such as scooters, bicycles, and others, research should be conducted in parallel to ensure the highest possible level of safety for users. There is also an increase in the number of people using bicycles and electric scooters, which develop significant speeds. The problem of accidents among users of classic and electric bicycles and scooters is evident, and post-accident injuries pose a serious challenge to medical practitioners. The literature is rich in statistical analyses of accidents among users of scooters and bicycles, but there are no studies where the behaviour of users of bicycles, scooters, etc. is analysed. The authors of this study set out to develop a measurement system to assess the traffic safety of people using bicycles and scooters. The device uses LIDAR to record the speed of the vehicle and a camera, the images of which are processed by an algorithm in order to classify the user as being on a bicycle or scooter and using or not using head protection with a helmet. It is also possible to analyse the behaviour of the vehicle users under study. The article describes the built measurement device and presents the results of the initial measurements made by the device.

**Keywords:** bikes and electric scooters safety; LIDAR measurement; bikes and electric scooters accidents; cycle paths safety; PLEV safety; PLEV's social sustainability

## 1. Introduction

Sustainable social development connected with the promotion of ecological transport and systematic development of widely understood personal transport influences the popularity of the use of bicycles, electric scooters, and other means of personal transport (PLEV), and requires the development of safe and functional transport infrastructure designed for those means of transport, which should reduce the risk of injuries to their users. Although bicycle infrastructure is included in the transportation policy of cities and agglomerations, including, among others, Warsaw [1,2], and the construction of cycling infrastructure is developing dynamically in Poland, so far there has been no country-wide planning and design document for safe infrastructure. As a result, many existing solutions do not meet the safety standards. The aspects of the technical standards for bicycle infrastructure contained in the Decree of the Minister of Transport and Maritime Economy on the conditions to be met by public roads and their location [3] have a very limited scope of regulation for this area. Some large Polish cities, including Warsaw [4], Poznan [5], Szczecin [6],

Wroclaw [7], Gdansk [8], Lodz [9], Kraków [10], and the Silesian Metropolis (Upper Silesian Metropolitan Union) [11], have developed their own technical standards in recent years, which often differ in their proposed functional and technical solutions. Most small and medium-sized towns and rural municipalities do not have such standards. There was a lack of a unified platform with safe design solutions, making it difficult to design safe cycling infrastructure. This directly translated into the safety of cyclists and PLEV users in the public area. In 2019, the Ministry of Infrastructure recommended the development of "Guidelines for the organisation of safe road traffic" [12] to be used in the preparation of investments. At the same time, it was emphasised that the study does not constitute technical and construction regulations within the meaning of the Act.

The cyclist as well as the PLEV user is a vulnerable road user. Like a pedestrian, he/she has no physical protection against a possible incident. In the bicycle/PLEV relationship, another vehicle, the bicycle/PLEV user, is weaker and more vulnerable in terms of risk to life and health. In a bicycle/PLEV–pedestrian relationship, the pedestrian is the weaker road user. The bicycle/PLEV travels at a higher speed than the pedestrian, which causes a measurable safety risk for pedestrians. A bicycle, according to the Traffic Law [13], should move on roads dedicated to it, a bicycle path, or a road for bicycles and pedestrians. If there is no such infrastructure, it should move on the carriageway or the shoulder. In special cases, according to the applicable regulations, it may move together with pedestrians on the pavement. The issue of electric scooters, personal transport devices, and assistive devices was regulated by law in Poland in March 2021 [14]. According to the amendment, the driver of a PLEV is obliged to ride, as is the case with bicycles, on a bicycle road or possibly a road for bicycles and pedestrians. The use of the pavement or footway by the driver of a PLEV is allowed exceptionally, that is, when there is no dedicated cycle path. In the case of persons using personal transport devices, they are obliged to use the pavement, the pedestrian way, or the cycle path where the right-hand traffic applies.

In Poland, bicyclists represented 7% of road fatalities per 1 million residents between 2018 and 2020, a share that was approximately 59% higher than the European average of 4.4% for those years [15] (in the CARE1 Community Database on Accidents on the Roads in Europe, data are available with a two-year delay). The danger to cyclists in Poland is one of the highest in Europe in terms of the number of cyclists killed per 1 million inhabitants (Figure 1).

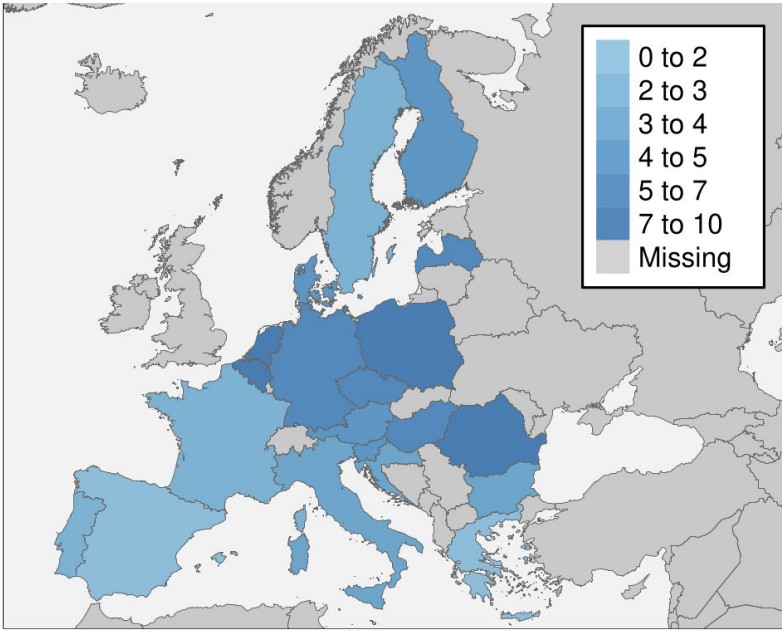

© EuroGeographics for the administrative boundaries

**Figure 1.** Number of cyclist road fatalities per 1 million inhabitants in the European Union countries in 2018–2020. Source: CARE European Road Accident Database, European Commission [15].

In Poland, cyclists accounted for 10% of the total number of road deaths between 2018 and 2020, a slightly higher share than the European average of 9% for those years [15]. The safety risk for cyclists in Poland is comparable to the European average in terms of the ratio of cyclists killed to the total number of those killed in road accidents (Figure 2).

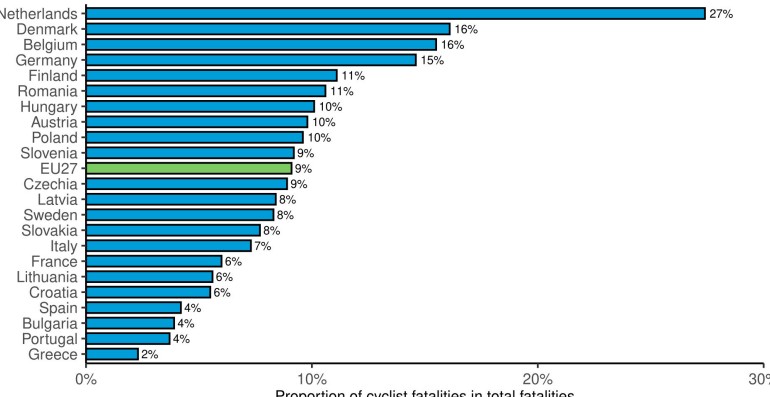

**Figure 2.** Percentage of cyclist fatalities among total fatalities by EU country 2018–2020 (The green colour indicates the EU average). Source: CARE European Road Accident Database, European Commission [15].

On the other hand, analysing in Poland the percentage change in the number of cyclist fatalities from 2011–2013 to 2018–2020, a decrease of −14% can be observed [15], and this is a good result in relation to the EU average of −1% (Figure 3). The reason for this is the improvement of road infrastructure, specifically the increase in the number of bicycle paths and roads, resulting in the separation of bicycle and car traffic.

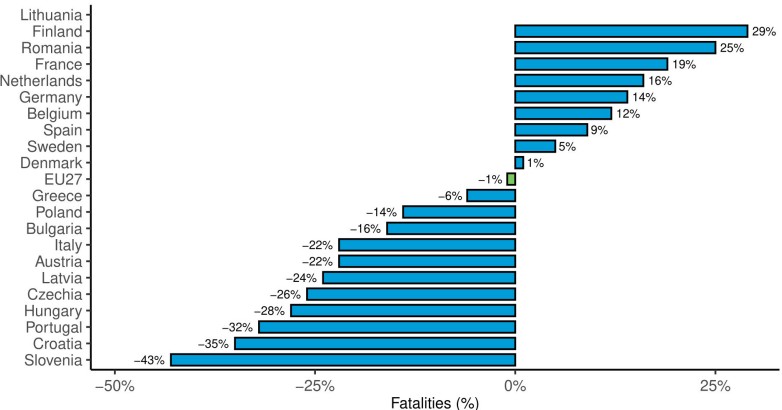

**Figure 3.** Percentage change in cyclist fatalities by country in the EU-27 (2018–2020 and 2011–2013) (The green colour indicates the EU average). Source: CARE European Road Accident Database, European Commission [15].

In Poland in 2015, the share of cyclists on the national road network in the overseas was 0.3% and on the provincial road network in the overseas was 2% [16]; in Warsaw in 2017 this share was 4.5% [17]. Although the popularity of bicycles in Poland is growing rapidly, it is still much less than in the Netherlands, where bicycles account for about 30% of travel. This means that, in relation to the kilometres travelled, the danger to cyclists in Poland is even greater.

Cyclists are considered vulnerable road users and are at high risk of serious injuries and death [18]. The most affected group among cyclists are elderly people over 60 years of age (Figure 4). Cyclists rank 4th in terms of the number of fatalities and serious injuries per 100 accidents, following motorcyclists, pedestrians, and moped riders (Figure 5).

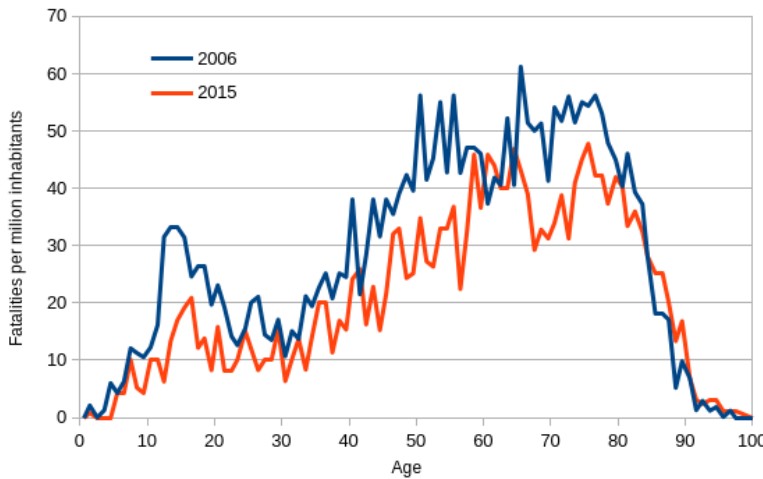

**Figure 4.** Road accident fatalities of cyclists by age in the European Union (data from May 2017). Source: CARE European Road Accident Database, European Commission [15].

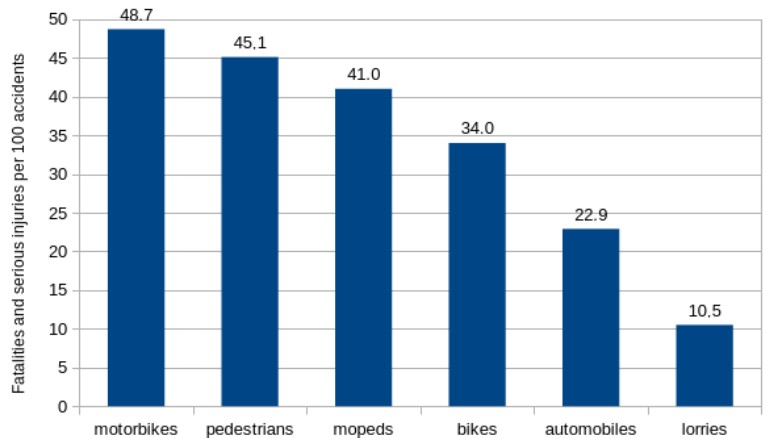

**Figure 5.** Accident severity by vehicle category in Poland in 2015–2017. Source: Polish Traffic Safety Observatory database.

The high risk for cyclists results from, among other things, the specifics of the vehicle [19], which is:

- Not very stable (especially at very low speeds);
- Not very resistant to lateral gusts of wind or blows of air caused by fast moving, large vehicles;
- Sensitive to unevenness and slipperiness of the road surface.

Furthermore, cyclists are deprived of any protection (apart from a helmet) against the effects of a fall or a collision with another vehicle or an obstacle. The high risk is also caused by the large difference in speed between cyclists, and other vehicles and the significant difference in weight.

Accidents involving cyclists can be caused by faults in the infrastructure, as well as the behaviour of cyclists and other road users [20]:

- The occurrence and severity of cycling accidents is influenced by both the quality and the general layout of the infrastructure. Poor quality of the road surface and its surroundings (e.g., breach, ditches, slopes) is often the cause of cycling accidents. Surface irregularities play a role in 6% accidents involving bicycles. Almost one-third of single accidents involving cyclists on road bikes occur on slippery road surfaces with longitudinal drainage.
- When cycling, there is a risk of falling due to the inherent instability of the vehicle. Safety can also be affected by the type of bicycle used, as different types require

different positions and have different ease with which you can place your feet on the ground. The width of the tyres is also a factor to consider. A safe bicycle should have a sturdy frame, reliable brakes and lighting, and well-profiled tyres. For people 65 years or older, maintaining balance while cycling can be challenging. Therefore, it is recommended that this age group use specially designed tricycles.

- Unsafe behaviour by road users, such as speeding, distracted driving, running red lights, or driving under the influence of alcohol, increases the risk of accidents, particularly those involving cyclists. Similarly, unsafe behaviour by cyclists themselves, such as using smartphones, cycling under the influence, or cycling without adequate lighting, also increases the risk of accidents. Accidents often occur due to a combination of unsafe behaviour by cyclists and other road users.

To take remedial action, it is very important to know the circumstances and causes of accidents. General figures collected by the police give us an idea of the scale of the risks but often do not explain precisely why specific accidents occur. To this end, in-depth investigations are carried out on the most serious accidents. Investigation teams are set up to travel to the scene of incidents immediately after they have occurred. As it is not always possible for the team to arrive at the scene of the accident quickly (long distance, heavy traffic, detours associated with the incident), a detailed analysis of the investigation documentation containing witness statements and precise information about the behaviour and injuries of the accident participants can be a source of additional data. Monitoring is another method of collecting data on inappropriate cyclist behaviour. The collected data can help determine the extent of inappropriate behaviour among traffic participants and identify the causes of unfavourable events. This information can be used to develop effective methods to combat these threats on the road. Such tasks were also carried out as part of the In-Depth understanding of accident causation for Vulnerable road users (InDeV) project funded by the European Union. The aim of the authors of this study was to develop a measuring apparatus to collect data on users' use of the cycle path, including traffic volumes, vehicle speeds (both average and instantaneous), vehicle types, and personal protective equipment used by users. The research results will be used to analyse traffic conditions in the cycling infrastructure and PLEV and to propose safety improvements for participants on cycling routes.

## 2. Methods for Monitoring Road Traffic Conditions

Devices, such as image recording equipment, are used on public roads to record traffic conditions and improve safety. Speed cameras, for instance, photograph vehicles whose drivers have committed a speeding offence. However, their disadvantage in collecting traffic data is that the vehicle must be travelling above a certain speed limit. This monitoring method does not allow the collection of a wide range of data for statistical purposes and the evaluation of road safety. To gather more information in the form of images, city monitors or cameras are installed in strategic locations along the roads. Advancements in video system technology have made it possible to extract more information from the input data provided. Algorithms employing machine learning and artificial intelligence are increasingly supporting these activities [21,22].

The development of numerous tools [23–25] has been driven by the widespread interest in image processing. This includes tools related to vehicles and their movement. Regardless of the recognition method employed, the principle of operation remains the same. The procedure for detecting objects begins by loading an image, typically obtained from surveillance cameras, as a recording, in the form of frames. Each frame is analysed separately. Depending on the method used, the algorithm may perform image processing or proceed directly to detection. Image processing may involve cutting out the background. On analysis of Figure 6, it is evident that the road is a constant element in each of the example images. If an additional element, such as a vehicle, appears in the frame, it can be extracted using the described procedure. The result of this procedure is presented in the

form of white outlines in Figure 6. Subsequently, the shape of the object is recognised on the basis of the learnt model, and the object in the image is determined [26].

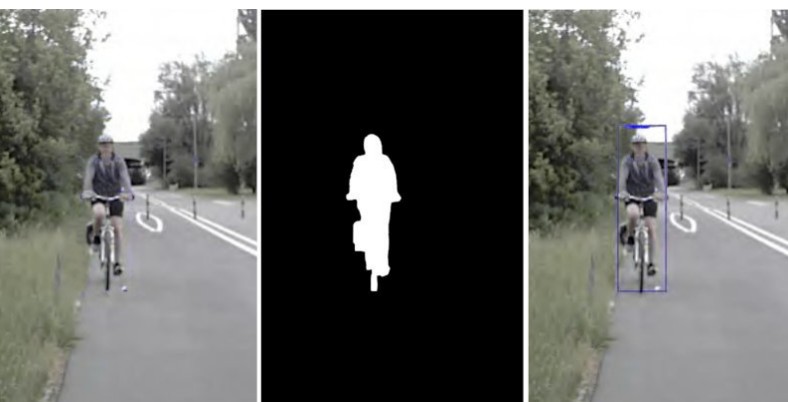

**Figure 6.** Stages of object detection [26].

Proper preparation of the model enables the detection of desired elements with greater precision and in more cases. One such model is the YOLO model (You Only Look Once), which is a convolutional neural network (CNN) commonly used for image analysis [27]. The training process begins with the preparation of a data set that demonstrates specific relationships. This data set comprises images and accompanying descriptions of the object positions to be detected. The images are divided into smaller objects to assign the appropriate weights. The neural network receives information about which of these objects contains the object being searched for and assigns a high weight to these areas. At this stage, attention is also paid to neighbouring areas and their relationship with each other [21]. On the basis of this data set, the model learns the conditions under which there is a probability that the object sought is present in a given location. By dividing the database into a learning part and a model checking part, it is possible to assess the extent to which the network correctly recognises the desired objects.

A properly equipped tool can detect various objects and their features. The authors' device enables the recognition of vehicle types without requiring detailed features. In addition to identifying basic vehicle types, the proposed detection tool can also recognise vehicle colours and speeds. The first involves determining the position of the centre of the surface of the detected vehicle in the image. A special script can be used to run a frame-by-frame analysis to assess speed. This method requires knowledge of the displacement and time, which allows for the calculation of average or instantaneous speed [28]. There are two methods to monitor the speed. The second method involves tracking the displacement of the number plate. The algorithm used in this case is similar to the one described earlier, with the exception that it analyses the image to detect vehicle number plates [29].

Object detection models are used for detecting vehicles, including motorbikes. They can detect the presence of a vehicle and determine whether the driver or passenger is wearing a helmet with a high degree of accuracy. This enables the collection of statistics on compliance with regulations and the enforcement of penalties for offences [21,30,31]. When analysing scientific articles, it was observed that models that are further versions of the YOLO model described above are used for object detection. This demonstrates that it is a suitable tool for handling the data acquired by the designed device. The presented methods offer numerous opportunities to analyse, assess, and improve safety among vulnerable vehicle users.

## 3. The Proposed Device Intended for Cycle and PLEV Traffic Monitoring

When designing the device, it was decided to split the data collection process into two stages. In the first stage, raw data were collected in the form of images and a file containing the distance of the object from the device, and the time when the distance was measured.

In the second stage, the collected data were processed on a computer using a proprietary script that:

- Loads data from the SD card;
- Sends photos to the neural network to detect objects in the photo;
- Calculates speed based on the distance of the object from the device and the time of measurement;
- Saves the processed object data (type of vehicle, personal protection used) and speed to a file.

The initial testing phase was conducted on a proprietary device constructed using standard modules. These modules were integrated into a dedicated housing that was specifically designed and manufactured for this project. To capture images for object analysis, an ArduCam Mini OV5642 camera (Figure 7(3)) was used, which is capable of capturing images with a resolution of 5 Mpx (2592 × 1944 px) and recording video at up to 120 fps. Communication with the camera module can be established through the SPI or I2C bus. The ArduCam module is compatible with microcontrollers such as the Arduino UNO, Arduino Mega, or Raspberry Pi Pico, and has configurable settings for taking pictures. Speed was measured indirectly using a Benewake TF03-100 laser distance sensor (refer to Figure 7(1)). This is point-based LIDAR that sends a signal in a narrow channel in the direction it is pointed. The chosen variant can measure distances ranging from 0.1 to 100 m with an accuracy of 0.1 m. Its field of view (FOV) ranges from 0.5° horizontally to 0.15° vertically. This range enables the detection of objects with a width of 0.87 m and a height of 0.26 m. The maximum refresh rate is 1000 Hz. Communication with the control unit can occur through the UART or CAN interfaces. Indirect speed measurement requires two quantities: time and distance. Time was measured using a module that allows for real-time tracking and counts down time using a quartz generator. A module built on the DS3231 integrated circuit with AT24C32 memory (Figure 7(2)) was chosen for its ability to provide battery backup for the process, independent of the power state of the microcontroller to which the RTC module is connected. This component enables tracking of the full date, including year, month, day, hour, minute, and second. Furthermore, a specialised library guarantees accurate representation of the number of days in a month or the tracking of leap years up to the year 2100. The RTC module communicates with the microcontroller through the I2C bus.

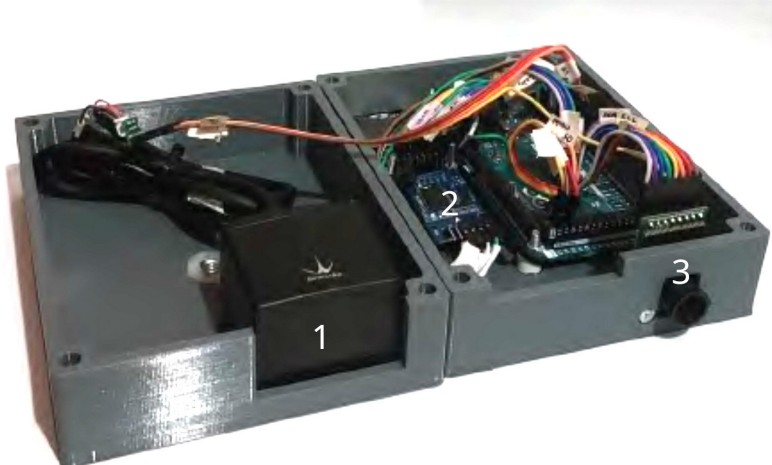

**Figure 7.** Construction of a bicycle traffic measuring device and PLEV (1—LIDAR TF03-100 laser distance sensor, 2—DS3231 RTC time module, and 3—ArduCam Mini OV5642).

## 4. Algorithms, Measurement Data Acquisition, and Control

The operating algorithm of the device is as follows. When the power is switched on, the LIDAR TF03-100 module sends a laser beam in the set direction, and the appearance

of an object at a distance of less than 100 m causes a text file to be created on the SD card and the following data to be recorded: the distance read and the reading time at the set interval. Based on these data, in a second step, the analysis computer script calculates the provisional speed of the vehicle between measurements. The number of intermediate speed measurements obtained depends on the frequency of the measurement (set in the software) and the length of time the object is in the sensor's field of view. When the object comes close enough for a photo to be taken, the data file (distance and time) is saved and closed, and the photo function is activated. The function reads the registers from the camera module and writes them to a file with the appropriate extension on the SD card. The name of both files has the following format '$year\_month\_day = hour + minute + second$' . This format allows the data processing to be automated and the date to be placed on the image in the correct format. The necessary date and time data are taken from the RTC module. By carrying out the above instructions, the necessary measurement data are stored on the SD card and the whole reading process is repeated for the next object to be studied. The use of the closed-loop algorithm allows continuous registration of vehicle traffic on the cycle path.

The second stage of the research involves the analysis of the acquired data, which are stored as image and text files on the SD card. When the SD card is inserted into the computer (SD reader), the script stored on the card recognises the files on the medium, groups them, and interprets the data provided. In the case of text files, the script calculates the temporary speeds and moves the file to the processed files folder. For image files, an image analysis is performed to detect objects in the image. It is important to identify the type of vehicle (car, bicycle, motorbike/scooter, electric scooter) and the personal protective equipment used by the riders, such as a helmet. If these elements are detected in the photo, they are marked with a rectangular outline and a comment. In addition, the program prints the date and the calculated average speed of the vehicle on the photo and then transfers the file to the appropriate folder.

The capabilities of the ImageAI library, which allows the manipulation of learnt models from a database of images, were used to recognise the objects in the image being analysed. The models are able to detect a range of objects with accuracy and precision, depending on their quality. The first model used was RetinaNet, which can detect 80 objects encountered on a daily basis. Due to the need to detect only vehicles, the model's scope was limited to bicycles, cars, and motorbikes.

Other modes of transport are also encountered on cycle paths. To demonstrate the possibilities of the project, a model was developed to detect scooters. This is the second of three methods used. The creation of files that allowed their detection was made possible by training the model with the library mentioned. This involves uploading a database consisting of images and a description of where the object to be detected is located. Depending on the size of the database, the model is more likely to learn correctly. The data set can be divided into a training area and a checking area. In the former case, the model has the task of learning the object detection methods, while the validation data allow the user to check how it performs in correctly detecting the object. Depending on the user's needs, the model can be trained many times until satisfactory quality is achieved.

The last object to be recognised, which is also the motivation behind the project, is a protective helmet. Its detection is performed by applying the last of the artificial intelligence object detection methods, using the roboflow library. It consists of sending the analysed image to the site where the trained model is located. There, again as in the previous cases, the image is analysed to detect the object being searched for. The purpose of using such a method was to demonstrate the possibilities associated with detecting objects from photographs. The advantage of such a solution is that there is no need for time-consuming model learning. In the case of scooter detection, the training took 100 h.

## 5. Device Testing and Results

The first device tests were carried out under laboratory conditions and consisted of verifying the correct functioning of the key components used in the traffic registration

system. A Leica Disto D8 laser rangefinder was used to verify the accuracy of the measured distances, and the test showed the correct functioning of the distance sensor used in the project. Another component of the system tested was the camera. The test demonstrated the requirement to use image capture as soon as the image was captured. The microcontroller used in the project in its basic configuration does not allow files to be saved directly to the computer's memory via the serial port, so a data carrier in the form of an SD card and a special library provided in the Arduino IDE environment were used. The above camera test also allowed the selection of image parameters such as resolution, saturation, brightness, contrast, angle, colour filters, exposure, and focus. This was done to obtain graphic material of a quality that would allow further processing of the photo. The last important element in the project, used to indirectly measure the speed of the object under study and to name the files (date, time), was the RTC real-time clock. Its correct operation was checked by comparing the readings with the time counted down in the computer, with measurements being taken at different time intervals. The tests carried out confirmed the correct operation of the system developed under laboratory conditions.

The next stage in verifying the correct operation of the device was to carry out tests under real conditions. This involved determining the range and type of data to be verified. The tests were carried out on two types of vehicles: a bicycle and an electric scooter. In the first stage, the vehicles were driven at a constant speed to allow comparison of the set values with those obtained by the device under test. The drivers of the vehicles involved in the test wore different headgear to verify the correct operation of the script used to detect objects in the image. The tests were carried out for three variants: without headgear, with a hat, and with a bicycle helmet.

The test site for the device was a cycle track, which allowed the experiment to be carried out with only one vehicle approaching the device. This eliminated measurement uncertainty caused by multiple objects appearing within the range of the LIDAR sensor. The tests were carried out in good weather conditions with good air clarity to ensure good illumination of the test vehicle. The experiment was carried out at a measurement distance of 10 m, with the test vehicle starting at a distance of 20 m from the instrument. It was found that the best results were obtained by aiming the laser sensor beam at the chest area of the person moving on the test vehicle. This configuration allowed the LIDAR sensor to improve the accuracy of the distance reading of the test subject.

The results obtained confirm the ability of the device to record parameters that are consistent with the actual condition. Figure 8a shows a measurement taken on a cycling path, where the test subject is a cyclist wearing a cycling helmet and moving on a bicycle. The device correctly detected all characteristics of the moving object. The average speed measured by the device was 15.36 km/h and differed by less than 1% from the average speed (15.25 km/h) calculated based on the time taken to cover the measurement distance, measured with a stopwatch.

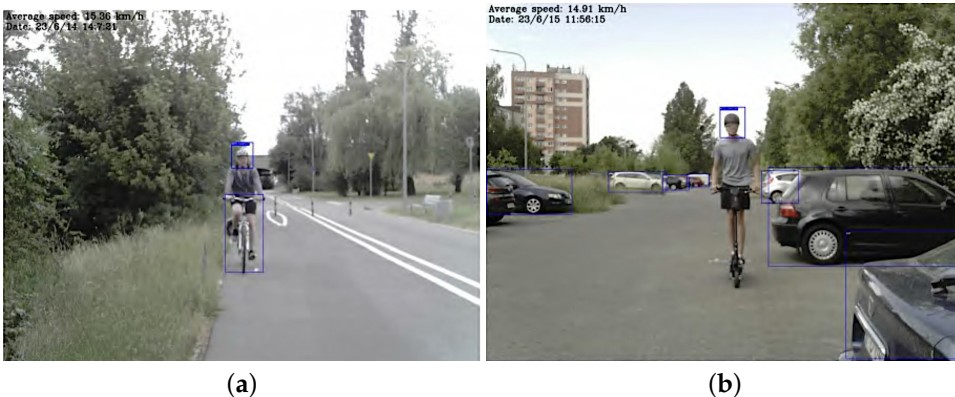

(**a**)  (**b**)

**Figure 8.** Graphical file after graphic analysis with information on average speed, date, and time of measurement: (**a**) bicycle, (**b**) scooter.

Figure 8b shows a measurement taken in a car park, where the subject is a person moving on an electric scooter wearing a bicycle helmet. The feature of the moving object being a bicycle helmet was correctly detected, but the problem was the detection of the electric scooter. This was most likely due to the fact that this object did not have the characteristic elements in the image taken from the front. The average speed was measured correctly, which in this case was 14.91 km/h, approximately 0.5% off the target speed of 15 km/h.

The field tests also showed the need to refine the device's control algorithm, as there was a problem taking the photo in about 30% of the cases. The problem was that the photo was taken too late, i.e., the photo was taken when the target was already out of frame. This was because the loop responsible for the instantaneous measurement of the object's speed took too long. The measurement method and the algorithm responsible for calculating the average speed should also be refined, as there were measurements in which the discrepancy between the measured and expected values was several km/h. The problem to be solved is related to the detection of scooter-type objects. The image processing algorithm detected this type of object only 15% of the time. Two measures can be taken to improve the detection performance. The first is to use a sufficiently large amount of data containing different orientations of the object in the image, and/or the second is to reconfigure the device so that, in the image taken, the object under study is visible from the side rather than the front. This will result in the photograph having more features, which will lead to more accurate object detection. The use of functionally separate devices that communicate with each other also seems to be a sensible solution. The appearance of an object triggers the process of photographing from the side, recognising the object, and at the same time triggering the examination of the object from the front.

## 6. Conclusions

The popularity of micromobility among people who move around urban areas on a daily basis continues to grow. More and more people are moving on bicycles, electric scooters, or monocycles, which are largely electric or electrically assisted vehicles. Vehicles of this type can develop significant speeds; the traffic law imposes restrictions on importers and sellers of such vehicles on the maximum speed or power of the electric motor used in these vehicles, but it is common to see vehicles on bike paths that do not meet these restrictions. Drivers of bicycles or PLEV vehicles are also capable of dangerous or reckless driving. The problem of accidents among users of these types of vehicles is evident, and post-accident injuries are often serious, especially among users who do not use helmet head protection. The literature is rich in statistical analyses of accidents among users of bicycles or PLEV vehicles, and now databases are beginning to be supplemented with accident statistics for relatively new vehicles such as electric scooters. There is a noticeable lack of studies in which the behaviour of users of bicycles and electric scooters/PLEV vehicles is analysed on the basis of data collected under real conditions in specific locations, where the behavior of users of given vehicles in given situations can provide a benchmark for evaluating both the causes of accidents and proposals for organisational changes resulting in improved safety.

The aim of the study was to develop a device that fills a gap in commonly used video surveillance and image processing equipment. The developed mobile device is independent of the city's surveillance infrastructure and can be easily moved between different measurement points where we are faced with an increased number of accidents. Initial tests of the proposed measuring device have confirmed its usefulness and practicality in capturing valuable experimental data on the speed of moving PLEV vehicles, the types of vehicle travelling on paths, and the use of bicycle helmets by users. The collected data makes it possible to assess user behaviour under well-defined real conditions. There is also some likelihood of recording dangerous situations and even accidents. Supplementing the device with the ability to take a series of images of a single object of sufficient quality can provide a reference point for researchers interested in both the degree of protection of,

for example, a helmet and the evaluation of the "aggressiveness" of environmental elements (kerbs, poles, barriers). The collected research material can be used in a comprehensive way, including for analyses related to possible traumatisation of specific body parts.

Further work on the device will include eliminating the problem of not registering the object under study in the frame, attempting to take a picture with the object edgeways to the camera, and attempts to take a series of pictures of that object. Work is also planned to "train" the measurement device using artificial neural network models to improve the accuracy of feature detection in the image. The modified device will be included in a research programme to evaluate the traffic conditions of users of PLEV vehicles, including modern electric scooters. We also plan to collect information forms the basis for further analytical analyses related to vehicle movement mechanics.

The method of speed measurement with the LIDAR sensor used in the device allowed measurement of one vehicle at a time. This was due to other objects being obscured by the test object. In the case of heavy traffic (small distances from the preceding vehicles), the solution is to place the measuring device at a certain height above the roadway level, to eliminate the vehicle being obscured by the preceding one. The angle between the device and the object under investigation should be included in the algorithm. However, this does not eliminate the problems associated with wide cycle paths and parallel moving objects. In this case, it is not possible to measure the speed of two objects moving in parallel with the LIDAR sensor. The only solution, which will also be tested, is to take a series of images with a camera and, on their basis, assess the speed of the moving objects. The LIDAR sensor will be used as an element to detect the appearance of objects within the camera's operating range to reduce the number of photos taken when there is no object in the area.

**Author Contributions:** Conceptualization, M.O., J.M. and J.A.; methodology, M.O., D.K. and J.M.; software, J.M.; validation, M.O., J.M. and D.K.; formal analysis, M.O. and J.A.; investigation, M.O., J.M., D.K. and J.A.; writing—original draft preparation, J.A. and J.M.; writing—review and editing, M.O. and D.K.; visualization, J.A.; supervision, M.O. and D.K.; project administration, M.O. and J.A. All authors have read and agreed to the published version of the manuscript.

**Funding:** This research received no external funding.

**Institutional Review Board Statement:** Not applicable.

**Informed Consent Statement:** Not applicable.

**Data Availability Statement:** The data presented in this study are available on request from the corresponding author.

**Conflicts of Interest:** The authors declare no conflicts of interest.

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
