# Peer review of "System for Monitoring the Safety and Movement Mechanics of Users of Bicycles and Electric Scooters in Real Conditions in the Context of Social Sustainability"

_sustainability, doi:10.3390/su16041684_

Round 1

Reviewer 1 Report

Comments and Suggestions for Authors

Review of the paper

"System for monitoring the safety and movement mechanics of users of bicycles and electric scooters in real conditions"

The article addresses the increasing popularity of bicycles and electric scooters, which have raised concerns about road safety. The authors identify a lack of studies that specifically analyze the behavior of users of these vehicles and the corresponding accidents and injuries. In response to this gap, they propose a measurement system using Lidar and a camera, coupled with an algorithm for classifying users and analyzing their behavior.

One of the key strengths highlighted in the article is the use of Lidar, which allows for the accurate measurement of vehicle speed, which can become a crucial aspect in assessing road safety for cyclists and electric scooter riders. Additionally, the camera and algorithm system provide insights into user behavior through recognition whether they are wearing a safety helmet or not.

Preliminary results indicate the effectiveness of the measurement device, although there are some challenges to be addressed. The identification of test objects, particularly individuals on electric scooters, needs refinement. The authors claim to focus on modifying the device and improving object detection in the future.

Overall, the article introduces a promising approach to monitor and evaluate road safety for bicycles and electric scooters using AI and sensor technologies. The use of Lidar and image processing algorithms demonstrates the potential to gather valuable data on user behavior and improve safety measures. The research contributes to the growing field of AI applications in road safety and could potentially have significant implications for reducing accidents and minimizing post-accident injuries among cyclists and electric scooter users. 

Together with this, the English language should be improved. It will be useful to double-check for typos/misspellings, grammatical errors, punctuation conventions, and letter cases; use spelling and grammar checkers. In addition, the introduction section could be reduced as it contains a lot of statistics indirectly related to the proposed system. 

Despite of the mentioned shortcomings, the paper is well structured and organized. There is the logical flow of ideas, as well as clarity of headings and subheadings in the paper. The coherence and logical progression of ideas ensure that each section builds upon the previous ones. The paper follows a standard format and adheres to the conventions of the field.

Considering these findings and the overall quality of the research, I would recommend this paper for publication after small revision.

Comments on the Quality of English Language

Together with this, the English language should be improved. It will be useful to double-check for typos/misspellings, grammatical errors, punctuation conventions, and letter cases; use spelling and grammar checkers. In addition, the introduction section could be reduced as it contains a lot of statistics indirectly related to the proposed system.

Reviewer 2 Report

Comments and Suggestions for Authors

The indicated use for monitoring the movement of cyclists and pedestrians is justified. However, when proposing a new device concept, its potential advantage over already used road traffic monitoring devices should also be demonstrated. For example, video recording and digital image processing techniques are widely used. In the case of cycling, extensive research has been carried out, including as part of the EU InDeV project. The article lacks comparisons and indications of potential advantages of the "new device".

The device description does not specify how objects moving on wide bicycle paths (by directing the LIDAR beam) or in the stream of bicycles (several objects appearing simultaneously on a given road section) are to be detected. There is also no information on how many objects the device can register in one measurement cycle to fill the capacity of the SD card. This information should be completed.

The development status of the described measuring device corresponds to the conceptual phase. The authors themselves indicate the need for many additional improvements to the presented device. They also provide data on the high unreliability of identifying electric scooters.

The combination of recording data about the movement of an object using a LIDAR device and image recording using a camera is valuable and very important in the aspect of the development of new measurement technologies. What remains to be solved is the synchronization of image acquisition in a very short time with the result of processing large data sets from the LIDAR device. The authors have not yet solved this problem satisfactorily. Therefore, it is difficult to indicate a new, significant contribution to the development of knowledge and new measurement techniques.

Other editorial comments.

In the introduction, the authors provide outdated information on bicycle infrastructure design standards in Poland - they quote an outdated document [3] and omit national bicycle path design standards introduced in 2023.

The statistical information on risk in bicycle traffic contained in the article does not introduce new elements of knowledge in relation to the main problem of the article, which is monitoring the movement of cyclists and electric scooters. The problem of security threats to the traffic of these user groups is widely known. It would be much more appropriate to present an overview of works devoted to monitoring the behavior of cyclists and other bicycle path users in real road traffic, together with practical conclusions arising from such monitoring. It would also be advisable to refer to the numerous works on surrogate measures of road traffic safety, as it explains which traffic parameters are important in proactive road safety management and which should be identified.

To sum up, the authors concluded that preliminary tests of the proposed measuring device confirmed its usefulness and practicality in recording valuable data on the movement of cyclists and other users of bicycle paths. This is not sufficient to confirm the need for further development of the described device. Authors should supplement applications with information to what extent the constructed device can replace video monitoring (cameras with digital image processing), which is already often used.

Reviewer 3 Report

Comments and Suggestions for Authors

I suggest explaining in more detail the algorithm used to recognise objects from photos, since one of the problems encountered is precisely the correct recognition of the scooter. It would also be useful to have a tabular presentation of the results of the measured speeds, also considering the distances to the object. 

Line 340 states that the system can also be used for "assessing the 'aggressiveness' of 

environmental elements" This should be better explained how this is possible.

Round 2

Reviewer 2 Report

Comments and Suggestions for Authors

I accept the authors' responses to previous comments. It is a pity, however, that the authors made too little use of these comments to supplement the text of the article with comments. For example, there is no comment on whether it is possible to develop the device in such a way as to record various objects on a wide bicycle path without the need to manually direct the recorder to the selected object. I expect that such a comment will be included in the summary.
